# Suitable Habitat Prediction and Analysis of *Dendrolimus houi* and Its Host *Cupressus funebris* in the Chinese Region

Guangting Miao †, Youjie Zhao †, Yijie Wang, Chunjiang Yu, Fei Xiong, Yongke Sun and Yong Cao *

School of Big Data and Intelligent Engineering, Southwest Forestry University, Kunming 650224, China; mgt@swfu.edu.cn (G.M.); zhaoyoujie@163.com (Y.Z.); wyj@swfu.edu.cn (Y.W.); ycj@swfu.edu.cn (C.Y.); arlequin@163.com (F.X.); sunyongke@swfu.edu.cn (Y.S.)
* Correspondence: cncaoyong@swfu.edu.cn
† These authors contributed equally to this work.

**Abstract:** The *Dendrolimus houi*, a phytophagous pest, displays a broad range of adaptations and often inflicts localized damage to its hosts. *Cupressus funebris*, an indigenous timber species in China, is significantly impacted by *D. houi*. Investigating the suitable habitat distribution and changes in *D. houi* and its host plant, *C. funebris*, within the context of climate warming, is essential for understanding *D. houi*'s development and providing novel insights for managing *D. houi* and conserving *C. funebris* resources. In this study, MaxEnt was employed to simulate the distribution of *D. houi* and its host plant, *C. funebris*, in their suitable habitats, evaluating the influence of environmental factors on their distribution and determining changes under a warming scenario. MaxEnt model parameters were adjusted using the Kuenm data package based on available distribution and climatic data. The minimum temperature of the coldest month emerged as the primary environmental factor influencing the distribution of a suitable habitat for *D. houi* and *C. funebris*, with a percentage contribution of environmental factors over 60%. There was a substantial similarity in the suitable habitat distributions of *D. houi* and *C. funebris*, with varying degrees of expansion in the total habitat area under different temporal and climatic scenarios. Intersection analysis results indicated that the 2041–2060 period, especially under low (SSP1-2.6) and high (SSP5-8.5) emission scenarios, is a critical phase for *D. houi* control. The habitat expansion of *D. houi* and *C. funebris* due to climate change was observed, with the distribution center of *D. houi* shifting northeast and that of *C. funebris* shifting northwest.

**Keywords:** MaxEnt model; environmental variable; *Dendrolimus houi*; *Cupressus funebris*; suitable habitat

## 1. Introduction

*Dendrolimus houi* Lajonquiere (*D. houi*) is an insect belonging to the genus Dendrolimus of the family Lasiocampidae in the order Lepidoptera, commonly known as dead leaf moths. It ranks among the top 100 insect pest species in terms of its geographical distribution [1]. This pest is widely prevalent in various regions of China, including Zhejiang, Fujian, Sichuan, Yunnan, Guangxi, Guangdong, Hunan, Guizhou, and other provinces [2]. *D. houi* primarily parasitizes trees such as *Cryptomeria japonica* and *Cupressus funebris (C. funebris)* and has a notable presence across an expansive area of 100,000 to 200,000 hm² [1]. *C. funebris*, as a key species for reforesting barren hills in southwestern China, plays a pivotal role in activities such as water conservation, soil preservation, and climate regulation. Studies have indicated that *C. funebris* is a particularly favorable host for *D. houi*, surpassing other potential hosts except for *C. japonica* [3]. *D. houi*, a broad-feeding phytophagous pest, exhibits robust adaptability across various host plants. Its larvae consume the needles and young leaves of *C. funebris*, thereby impeding the growth of *C. funebris* plants. This extensive feeding negatively impacts the growth rate of *C. funebris*, resulting in substantial economic losses and ecological damage. These detrimental effects place significant constraints on the production and development of *C. funebris*. In areas where infestation occurs, hosts often





suffer damage in patches. The impact ranges from the loss of needles and shoots in milder cases to the consumption of entire branches in severe instances. Current research on *D. houi* predominantly focuses on biological control and monitoring [4,5], with less attention directed towards understanding how *D. houi* and its host plants may respond to future climate changes.

The climate plays a pivotal role in shaping species distribution, population dynamics, and biological interactions [6–8]. As greenhouse gas concentrations continue to rise, the resulting increase in average surface temperatures is profoundly impacting global biodiversity [9]. Many species are rapidly responding to these climate changes [10]. Over the past century, numerous observations have confirmed significant alterations in Earth's climate, primarily characterized by global warming. The Intergovernmental Panel on Climate Change (IPCC) stated in its Fifth and Sixth Assessment Reports that global surface temperatures have risen by approximately 1 °C since 1850–1900. It further projects an average surface temperature increase of 0.3 °C to 4.5 °C by 2100. These assessments also highlight that climate change trends in China align with global patterns [11,12]. Climate change is diminishing the habitat suitability for various species, resulting in shifts in plant ranges. Lele Lin et al. conducted a study on 12 pine species within the *Pinus* genus, demonstrating that factors such as temperature and precipitation play a pivotal role in expanding or contracting the suitable habitats for *Pinus* [13]. Notably, temperature emerges as the most significant environmental factor affecting the population dynamics of insects [14]. Global warming may exacerbate pest issues and potentially expand the geographical range of insect pests. Yan Y et al.'s species distribution modeling of 76 pest species revealed that climate change could lead to an expansion in pest population distributions [15]. These findings underscore the profound influence of climatic factors on the range of pests and their host plants, emphasizing the urgency of predicting trends in both plants and the pests and diseases that affect them.

Phillips et al. pioneered the development of MaxEnt software, which features a self-checking function that autonomously generates ROC curves, Jacknife test results, and assessments of environmental factor contributions. The MaxEnt model, based on maximum entropy, predicts species distribution within a study area using known data points of species occurrence (>5) along with corresponding environmental variables [16]. This data-driven approach utilizes collected distribution points to analyze the ecological preferences of a species [17], quantified as a probability representing the species' habitat preference. MaxEnt models have become increasingly prevalent in recent years and find widespread application in various fields, including pest and disease control, the assessment of species invasions [18], medicinal plant studies [19], and the conservation of endangered species [20,21]. One of the key advantages of the MaxEnt model is its ability to yield accurate predictions even when species distribution information is incomplete and when correlations between climate and environmental factors are unclear. This makes it a valuable tool for species distribution modeling and habitat suitability assessment.

Drawing from the existing distribution data of *D. houi* and *C. funebris*, this study employs the MaxEnt model to explore both the present and prospective habitat suitability and the primary environmental factors affecting *D. houi* and its host, *C. funebris*. The investigation aims to anticipate alterations in their distribution patterns under scenarios of climate change. The study seeks to offer insights into the effective management of *D. houi* in China and to provide a scientific foundation for the preservation, introduction, and expansion of *C. funebris* resources.

## 2. Materials and Methods

This study is structured into three main steps. The initial step involves gathering data points and environmental factors related to species occurrence. Effective occurrence points are selected, and highly correlated environmental factors are eliminated. Optimal parameter combinations are determined using the Kuenm data package [22], and the chosen parameters are incorporated into model construction.

Moving to the second step, the model construction phase begins. The selected effective distribution points and environmental factors, following the screening process, are input into the model. Seventy-five percent of these distribution points are designated as the training set, with the remaining 25% allocated as the test set. The process is repeated ten times, with parameters set to the combinations exhibiting the smallest AIC values, as identified in the previous step.

In the third step, analyze Maxent output to determine model simulation capabilities and key environmental factors. The layers generated by the MaxEnt model are converted using ArcGIS [23] to perform delineation of current and future fitness zones, analysis of distribution changes, analysis of fitness zone centers, and intersection analysis. The full workflow on which analyses were based is summarized in Figure 1.

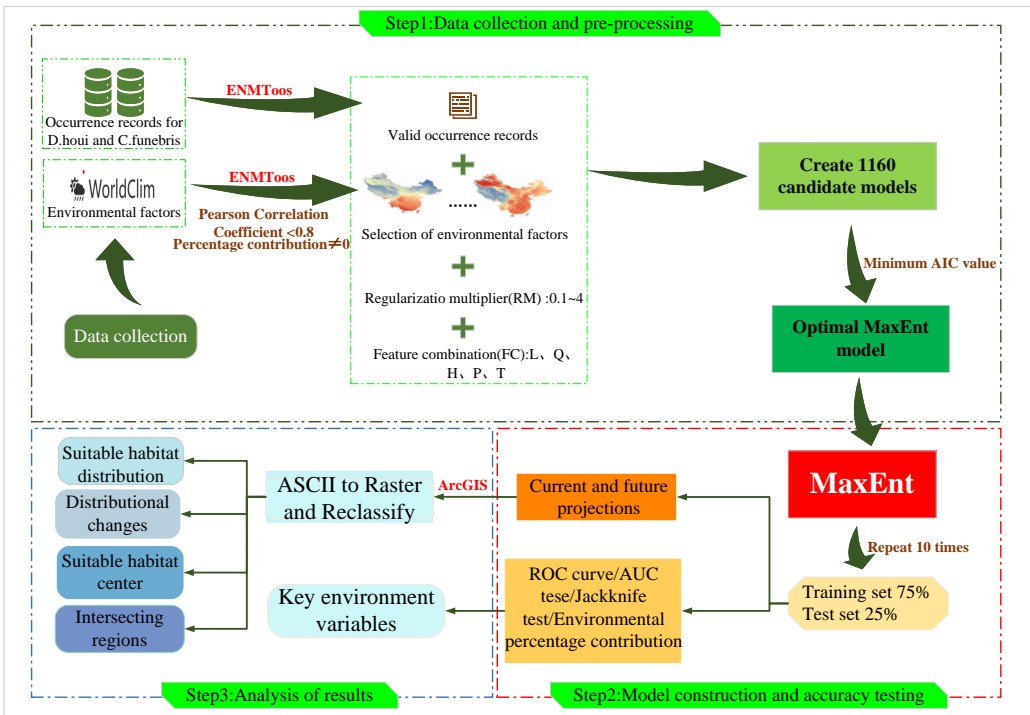

**Figure 1.** Workflow of this study. This research is divided into three steps. The first step is to conduct data collection and pre-training, the second step is to carry out model construction and accuracy testing, the third step involves results analysis.

### 2.1. Data on the Geographical Distribution of Species

This study collected geographical distribution data of *D. houi* from the China Animal Theme Database (http://zoology.especies.cn/ accessed on 26 April 2023), the Global Biodiversity Information Facility (https://www.gbif.org/ accessed on 26 April 2023), and CNKI (https://www.cnki.net/ accessed on 26 April 2023). Geographic distribution data for *C. funebris* was collected from the Global Biodiversity Information Facility (https://www.gbif.org/ accessed on 30 April 2023), National Plant Specimen Resource Center (https://www.cvh.ac.cn/ accessed on 30 April 2023), National Specimen Information Infrastructure (http://www.nsii.org.cn/2017/home.php accessed on 30 April 2023). The latitude and longitude of the distribution data were supplemented by the Baidu coordinate picking system, from which duplicates and distribution points with unclear information were eliminated. The geographical distribution data for *D. houi* and *C. funebris* were processed using ENMTools [24]. This tool can automatically match the grid size of environmental factors, eliminate redundant data within the same raster, and ensure the retention of only one valid distribution point per raster. The environmental factor data in this study utilized a grid size of 5 km × 5 km. Following the screening process resulted in

a dataset of 57 valid distribution points for *D. houi* and 348 valid distribution points for *C. funebris* (Figure 2).

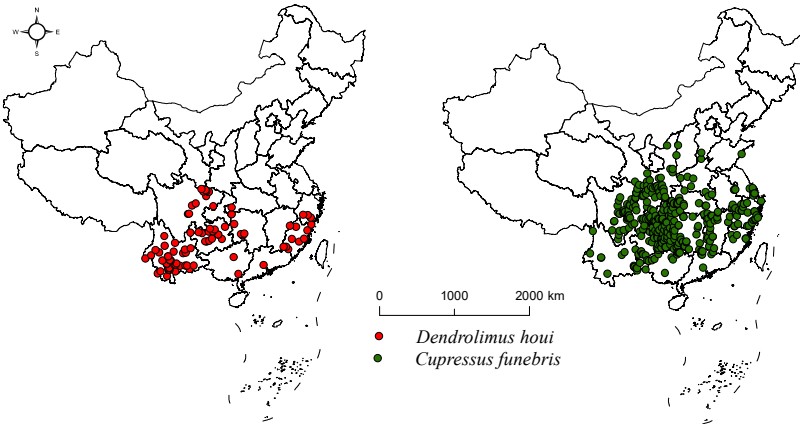

**Figure 2.** Geographical distribution of *D. houi* and *C. funebris*.

### 2.2. Environmental Data

Climate and elevation data were obtained from the World Clim database (https://worldclim.org/ accessed on 29 September 2023), and future climate data were simulated using the World Clim v2.1 data, which uses the Global Climate System model of the International Coupled Model Comparison Programme Phase 6 (CMIP6) to simulate future climate change. Future Climate [25] has chosen the sustainable development pathway SSP1-2.6, the representative concentration pathway SSP2-4.5, and the conventional fossil fuel-dominated pathway SSP5-8.5. The BCC-CSM2-MR model proposed by the Beijing Climate Centre of the China Meteorological Administration has a good simulation performance in East Asia [26], the BCC-CSM2-MR model data were therefore selected to simulate the distribution range of *D. houi* and *C. funebris*.

### 2.3. Environmental Factors Pre-Treatment

Handling multicollinearity in species distribution modeling is crucial to ensuring the reliability of the model's results [27]. Pearson correlation tests the environmental variables using ENMTools; this helps identify which variables are highly correlated with each other, and the results of the Pearson correlation test are shown in Figure 3. The model was constructed using the MaxEnt default parameters and the full range of environmental factors to obtain the percentage contribution of all environmental factors, excluding environmental factors with a contribution of 0. These variables may not significantly impact the model's predictions; retained variables have a correlation coefficient of less than 0.8, and this threshold helps ensure that only moderately correlated variables are retained. When the absolute value of the correlation coefficient of the environmental variables exceeds 0.8, the most suitable variable is chosen, since highly correlated variables can lead to multi-collinearity issues. Screening through the above steps resulted in the selection of seven environmental variables for subsequent studies (Table 1). This enhances the model's ability to accurately predict a suitable habitat and reduces the risk of multicollinearity, affecting the model's performance.

### 2.4. Model Construction and Evaluation

Initially, the model was constructed using the default parameters of MaxEnt and included all environmental factors. Subsequently, the environmental factors with significant contribution rates were identified and chosen for participation in the model optimization process. The regularization multiplier (RM) and feature combination (FC) parameters of the MaxEnt model were then adjusted by utilizing the Kuenm package. The model's complexity was compared under various parameter combinations, and the parameters yielding the

lowest complexity were selected for further model optimization [28]. The complexity of the MaxEnt model is significantly influenced by the regularization multiplier (RM) and feature combination (FC) parameters. The RM is used to control the complexity of the model and prevent overfitting. It introduces a penalty term in the log-likelihood function to balance the model's fit to the training data with the model's complexity. MaxEnt currently incorporates five FCs: linear (L), quadratic (Q), hinge (H), product (P), and threshold (T). Products are products of all possible pair-wise combinations of covariates, allowing simple interactions to be fitted. Threshold features allow a "step" in the fitted function; hinge features are similar, except they allow for a change in the gradient of the response [29]. The Kuenm data package assesses the complexity of the model through the AIC (AICc) values of the MaxEnt model, corrected for different parameter conditions. The AIC (Akaike Information Criterion) informativeness criterion serves as a metric for evaluating the goodness of the model fit, it involves balancing the complexity of the estimated model against the quality of the data used for fitting, the AIC informativeness criterion favors models with the smallest AIC values [30]. AIC can be expressed by Equation (1) [31]:

$$AIC = 2k - \ln(L), \tag{1}$$

where $k$ is the number of parameters and $L$ is the likelihood function. This criterion balances model simplicity (smaller $k$) with accuracy (larger $L$). In the context of model selection, where there are multiple candidate models (n), the AIC values for all models can be computed simultaneously. The model associated with the minimum AIC value is then prioritized for selection.

**Table 1.** Environmental factors selected for this study.

| Type | Environmental Factors | Description of Environmental Factors |
|---|---|---|
| Description of Environmental Factors | Bio1 | Annual Mean Temperature |
| | Bio2 | Mean Diurnal Range |
| | Bio3 * | Isothermality |
| | Bio4 * | Temperature Seasonality |
| | Bio5 * | Max Temperature of Warmest Month |
| | Bio6 * | Min Temperature of Coldest Month |
| | Bio7 | Temperature Annual Range |
| | Bio8 | Mean Temperature of Wettest Quarter |
| | Bio9 | Mean Temperature of Driest Quarter |
| | Bio10 | Mean Temperature of Warmest Quarter |
| | Bio11 | Mean Temperature of Coldest Quarter |
| Precipitation | Bio12 * | Annual Precipitation |
| | Bio13 | Precipitation of Wettest Month |
| | Bio14 | Precipitation of Driest Month |
| | Bio15 * | Precipitation Seasonality |
| | Bio16 | Precipitation of Wettest Quarter |
| | Bio17 | Precipitation of Driest Quarter |
| | Bio18 | Precipitation of Warmest Quarter |
| | Bio19 | Precipitation of Coldest Quarter |
| Terrain | elev * | Elevation |

The environmental factors labeled with * are the environmental factors for which the screening was completed for subsequent studies. Bio4: Temperature Seasonality (Temperature Seasonality is the ratio of the monthly average temperature to the standard deviation of the monthly average temperature. An increase in the Temperature Seasonality indicates a gradual increase in temperature difference).

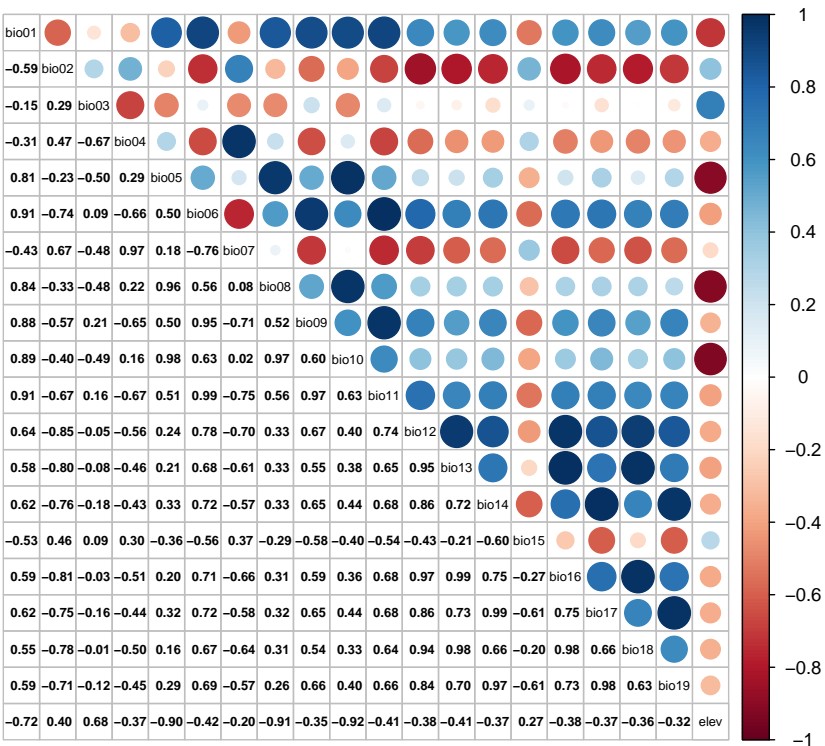

**Figure 3.** Matrix of environmental factor correlation coefficients. Larger circles in the graph indicate higher correlation and smaller circles indicate lower correlation.

In this study, the Kuenm packet selects the model with the smallest delta AiCc value as the optimal solution from 1160 models created with parameter settings involving 40 regularization multipliers [0.1–4] and 29 feature combinations [L, Q, P, T, H, LQ, LP, LT, LH, QP, QT, QH, PT, PH, TH, LQP, LQT, LQH, LPT, LPH, QPT, QPH, QTH, QTH, LQPT, LQPH, LQTH, LPTH, LQPTH].

The distribution point data and environmental factor data for *D. houi* and *C. funebris* were meticulously screened and imported into the MaxEnt model software. To set the RM and FC for the MaxEnt model when the model complexity is lowest, with 75% of the distribution point data randomly chosen for modeling and the remaining 25% for model validation. The number of model iterations was set to ten, and the average of these iterations was taken to demonstrate the model's simulation capability. The iteration run type was set to "Subsample", and default parameters were used for all other settings. Subsample refers to randomly selecting a subset of samples from the original dataset for training. The purpose of this process is to enhance training speed, reduce computational costs, and prevent overfitting. The Receiver Operating Characteristic (ROC) curve, along with the Area Under the Curve (AUC) value, is effectively used to evaluate the precision and effectiveness of a model. AUC values typically range from 0.5 to 1. A model with an AUC value of around 0.5 indicates poor predictive performance, A model with an AUC value between 0.7 and 0.9 is considered to have good predictive effectiveness. A model with an AUC value greater than 0.9 is considered excellent, indicating high accuracy in predictions. The relative contribution of each environmental factor to the model was evaluated according to the percentage contribution of the environmental factors output from the MaxEnt model, and the dominant environmental factors affecting the geographical distribution of *D. houi* and its hosts were selected. The output of the MaxEnt model is a raster layer in ASC format, and the default adaptive index interval is [0, 1]. The adaptive index p is obtained by the natural breakpoint grading method, and the threshold of the adaptive zone is determined by this method. The adaptive zones of *D. houi* and *C. funebris* are reclassified by ArcGIS, and the adaptive zones are divided into four grades: unsuitable habitat (0–0.1), poorly

suitable habitat (0.1–0.3), moderately suitable habitat (0.3–0.5), and highly suitable habitat (0.5–P).

### 2.5. Analysis of Results

The MaxEnt model was utilized to simulate the present and future distribution of *D. houi* and *C. funebris* under different climate scenarios, including SSP1-2.6, SSP2-4.5, and SSP5-8.5. This involved mapping the geographic distribution for the current, 2050s (2041–2060), and 2070s (2061–2080) time periods. The Spatial Distribution Modeling (SDM) tool within the ArcGIS software was employed to simulate changes in the suitable habitat distribution of *D. houi* and *C. funebris* under various climate scenarios, along with the shifting of their habitat centers. This was accomplished by overlaying the current and three future climate scenario layers to assess changes in the suitable habitat distribution for both species. Migration routes of suitable habitat centers for *D. houi* and *C. funebris* were simulated, taking into consideration different time frames and climate scenarios. To compare and analyze differences, the intersection analysis tool in ArcGIS was used after converting all data formats of the plotted layers.

## 3. Result

### 3.1. Model Optimisation and Accuracy

By default, the MaxEnt model is set with RM = 1 and the feature combinations include linear (L), quadratic (Q), product (P), and threshold (T). The optimized model achieved the delta.AICc value of 0, which is the most likely to be the optimal model. This indicates that the optimized model has the lowest complexity and provides the best fit for the data (Table 2). For the *D. houi* optimization model, the smallest delta.AICc value was obtained with RM = 3.4 and an FC of Q, T, H. In the case of the *C. funebris* optimization model, the smallest delta.AICc value was achieved with RM = 0.5 and an FC of L, Q.

**Table 2.** Optimizing evaluation metrics for MaxEnt models using Kuenm packets.

| Species | Type | RM | FC | delta.AICc |
|---|---|---|---|---|
| *Dendrolimus houi* | Default | 1 | LQPH | 269.46 |
| | Optimisation | 3.4 | QTH | 0 |
| *Cupressus funebris* | Default | 1 | LQPH | 132.67 |
| | Optimization | 0.5 | LQ | 0 |

RM: Regularization Multiplier; FC: Feature Combination; L: linear; Q: quadratic; P: product; T: threshold; H: hinge; AICc: Akaike Information Criterion; delta.AICc: Differences between AICc values of different models, a smaller delta. AICc suggests a higher probability that the corresponding model is the best fit.

The accuracy of the MaxEnt model was assessed using ROC curves. Figure 4 illustrates the mean training AUCs (Area Under the Curve) for the predicted *D. houi* and *C. funebris* models. The mean of the AUCs, calculated over 10 training iterations for both species, exceeded 0.9. This outcome suggests that the model's level of fit is highly accurate, and it can effectively simulate the suitable habitat distribution for *D. houi* and *C. funebris*.

### 3.2. Critical Environmental Factors

In the simulated suitable habitat models for *D. houi*, the percentage contribution, which measures the predicted impact of environmental factors on the species' fitness zone, was highest for the Min Temperature of Coldest Month (BIO6), Temperature Seasonality (BIO4), and Isothermality (BIO3), with a combined contribution of 94.3%. These factors played a crucial role in constructing the model. For the simulated suitable habitat model of *C. funebris*, the Min Temperature of Coldest Month (BIO6) made the most significant contribution, accounting for 91.2% of the total contribution (refer to Table 3). These environmental

variables were the primary factors in model construction. Results from the Jackknife method test of the MaxEnt model (refer to Figure 5) indicate that the Min Temperature of Coldest Month (BIO6), Annual Precipitation (BIO12), and the Temperature Seasonality (BIO4) had the highest impact on model performance in simulating the suitable habitat for both *D. houi* and *C. funebris*. This suggests that these three environmental factors alone contain the most valuable climatic information compared to other factors.

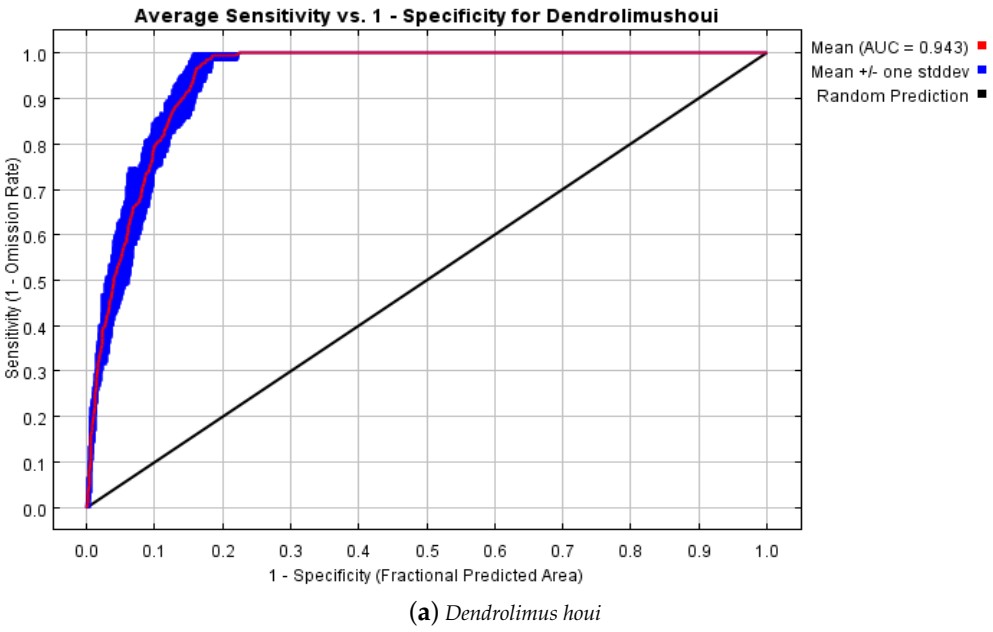

(**a**) *Dendrolimus houi*

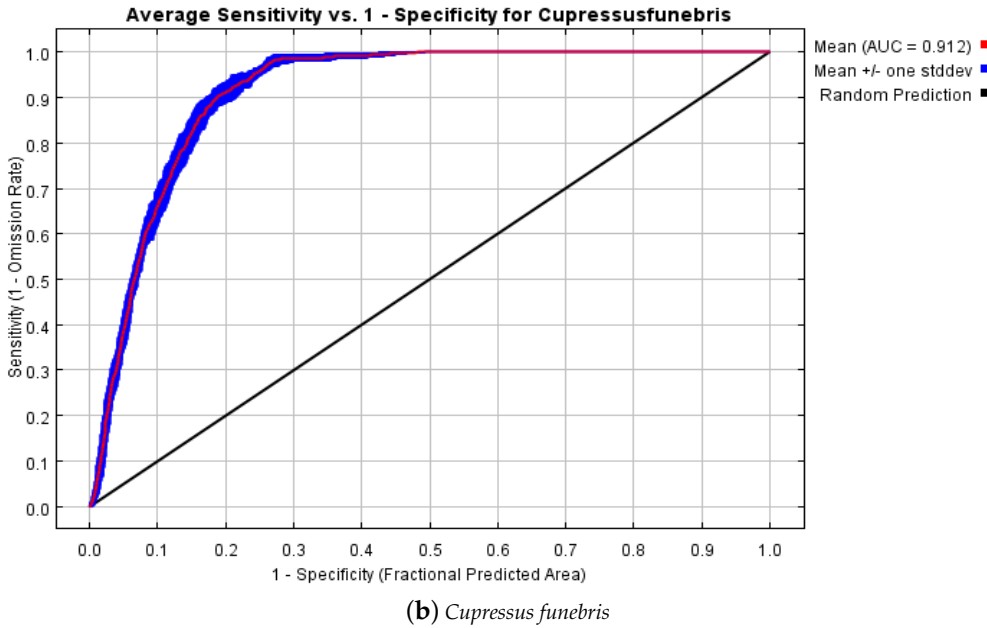

(**b**) *Cupressus funebris*

**Figure 4.** ROC curves for suitable habitat simulation models of (**a**) *D. houi* and (**b**) *C. funebris*.

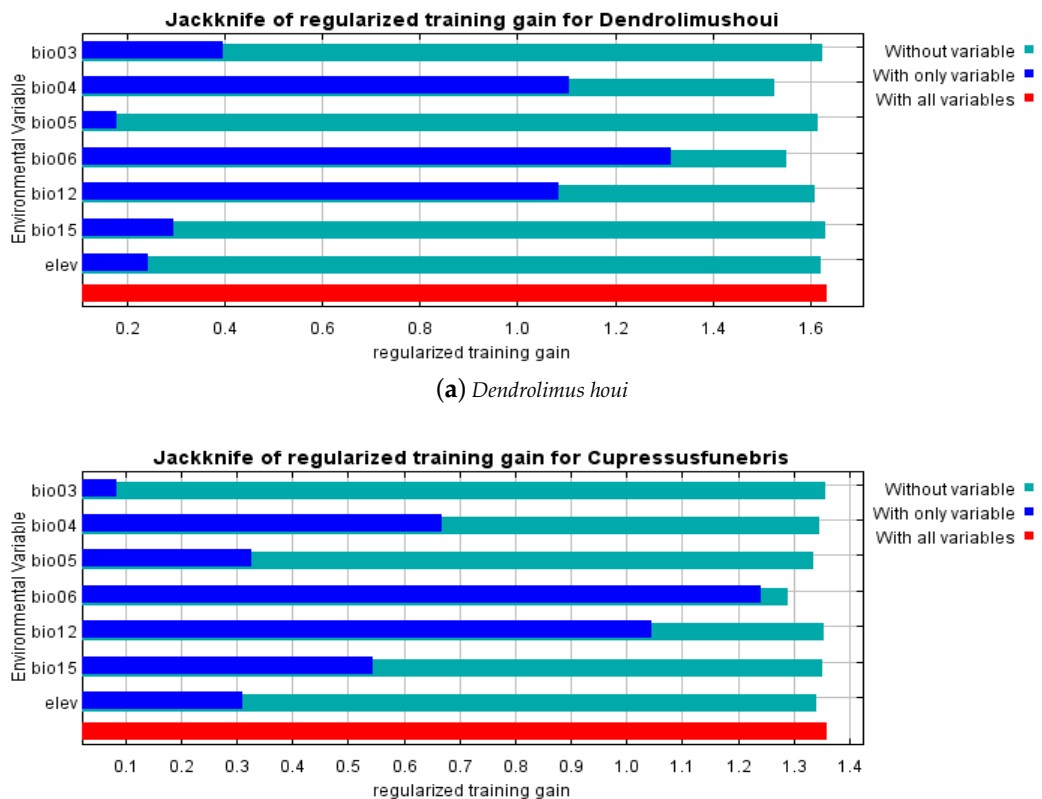

(**a**) *Dendrolimus houi*

(**b**) *Cupressus funebris*

**Figure 5.** Results of the Jackknife test for environmental variables (**a**) *D. houi* (**b**) *C. funebris*. bio3: Isothermality; bio4: Temperature Seasonality (Temperature Seasonality is the ratio of the monthly average temperature to the standard deviation of the monthly average temperature. An increase in the Temperature Seasonality indicates a gradual increase in temperature difference.); bio5: Max Temperature of Warmest Month; bio6: Min Temperature of Coldest Month; bio12: Annual Precipitation; bio15: Precipitation Seasonality; elev: Elevation.

**Table 3.** Contribution of each environmental factor to the simulated species distribution model.

| Variable | Environmental Variable | Perecent Contribution | |
| --- | --- | --- | --- |
| | | *Dendrolimus houi* | *Cupressus funebris* |
| Bio3 | Isothermality | **4.5** | **3.1** |
| Bio4 | Temperature Seasonality | **24** | **1.8** |
| Bio5 | Max Temperature of Warmest Month | 0.7 | 1.7 |
| Bio6 | Min Temperature of Coldest Month | **65.8** | **91.2** |
| Bio12 | Annual Precipitation | 3 | 0.4 |
| Bio15 | Precipitation Seasonality | 0.1 | 0.4 |
| Elev | Elevation | 2 | 1.4 |

### 3.3. Suitable Habitat for D. houi and Its Host under Current Climate Scenarios

MaxEnt predictions indicate the current geographical distribution of *D. houi* and *C. funebris*, as depicted in Figure 6. The total suitable habitat area for *D. houi* is approximately $236.82 \times 10^4$ km$^2$, with a highly suitable habitat covering $55.92 \times 10^4$ km$^2$. For *C. funebris*, the total suitable habitat area is about $262.97 \times 10^4$ km$^2$, and the highly suitable habitat encompasses $88.83 \times 10^4$ km$^2$. These suitable habitats are primarily located in Tibet,

Yunnan, Sichuan, Chongqing, Guizhou, Guangxi, Guangdong, Fujian, Hunan, Jiangxi, and Zhejiang. The combined total suitable habitat for both species constitutes 24.67% and 27.39% of the total land area in China, respectively. Furthermore, the highly suitable habitat represents 23.61% and 33.78% of the total suitable habitat area. The consistency between the suitable habitat distribution and existing literature records indicates that the model developed in this study effectively simulates the suitable habitat distribution of *D. houi* and *C. funebris*.

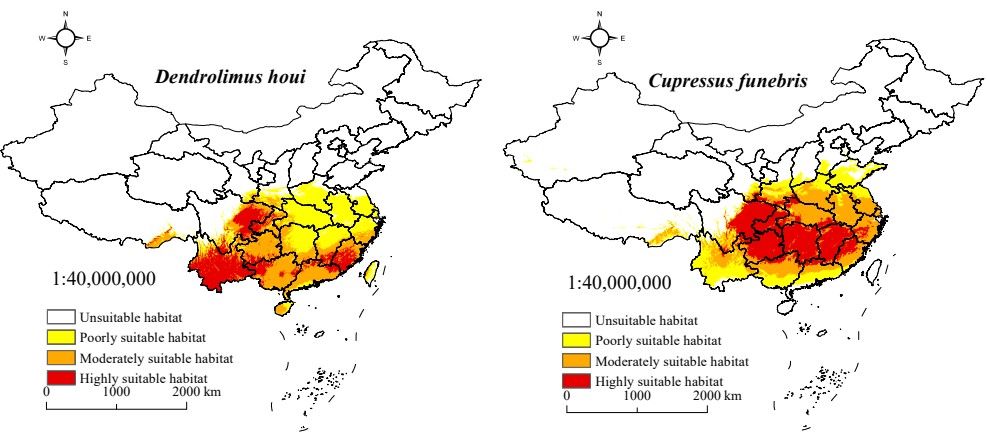

**Figure 6.** Distribution of suitable habitat for *D. houi* and *C. funebris* during the present period.

### 3.4. Suitable Habitat for D. houi and Its Host under Future Climate Scenarios

Under the three future climate scenarios, the regions with the most overlapping highly suitable habitats for *D. houi* and *C. funebris* are situated to the south of the Qinling-Huaihe River. While the highly suitable habitat for *D. houi* remains relatively stable, the overall suitability area extends further north of the Qinling-Huaihe River. Notably, the low suitability area ($0.1 < p < 0.3$) is projected to expand into Xinjiang for the first time in the 2070s. The highly suitable habitat for *C. funebris* expands over time, correlating with increasing greenhouse gas emissions. By the 2050s, the highly suitable habitat for *C. funebris* nearly encompasses the entire southern part of China and continues its expansion towards the north and northwest. Remarkably, in the 2050s, the highly suitable habitat for *C. funebris* is expected to make its inaugural appearance in Xinjiang ($p > 0.5$) (Figure 7).

Future climate change is influencing the distribution of *D. houi* and *C. funebris*, with the total suitable habitat area for both species showing a tendency to increase across all periods and concentration pathways under future climate scenarios, compared to the current period. The most favorable climate scenario for *D. houi* is SSP5-8.5, where the total suitable habitat reaches $287.26 \times 10^4$ km$^2$ in the 2070s, reflecting a 21.30% increase compared to the current period. The area of highly suitable habitat, on the other hand, was highest in the SSP1-2.6 scenario in the 2050s, reaching $91.83 \times 10^4$ km$^2$. For *C. funebris*, the SSP5-8.5 climate scenario stands out as the most suitable in the 2070s, with the total suitable habitat area expanding to $426.17 \times 10^4$ km$^2$, indicating a substantial 62.06% increase compared to the current period. The highly suitable area covers an area only $\times 10^4$ km$^2$ less than the total area of the current suitable habitat area (Figure 8).

The suitable habitat of *D. houi* remains stable while expanding northwards in response to climate change, with contraction areas identified in Shaanxi, Gansu, and Taiwan provinces. Notably, the contraction regions are smaller in size compared to the expanses of the newly established areas. In contrast, the potential habitat distribution for *C. funebris* is more dynamic. The contracted area is situated at lower latitudes along the southern edge of the overall suitable region, encompassing Yunnan, Guangxi, and Guangdong provinces. There are significant changes in expansion areas, stretching extensively from west to east across nearly the entire Chinese region (Figure 9).

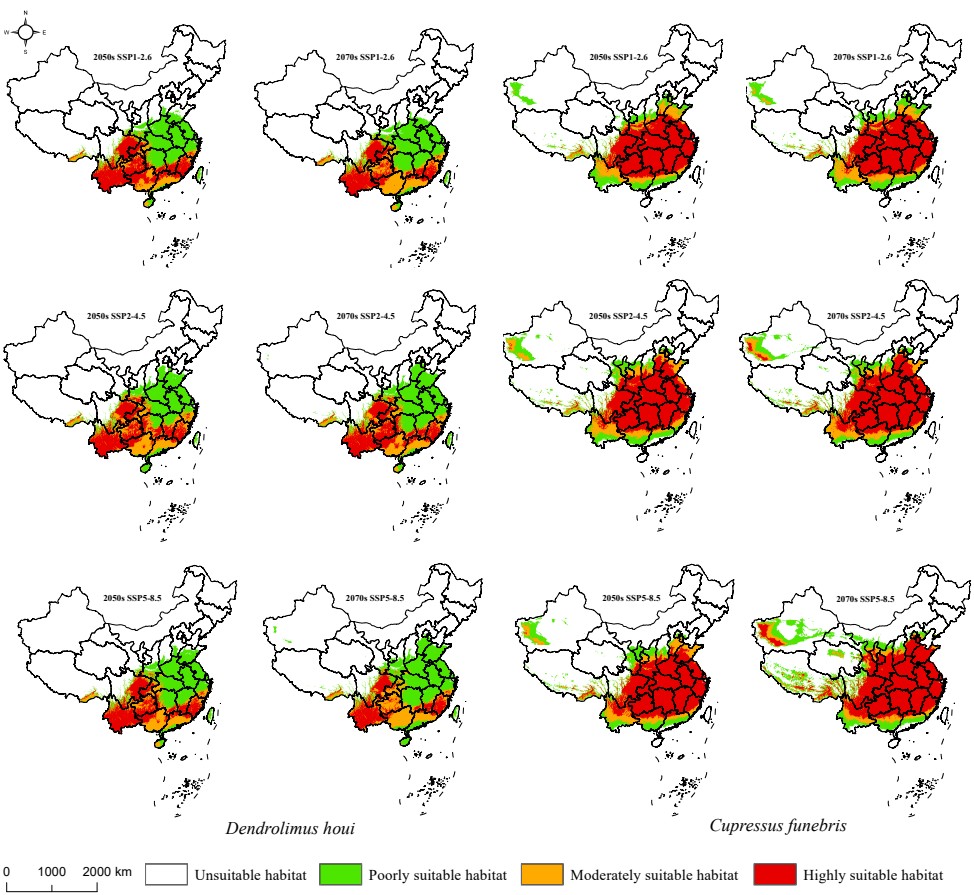

**Figure 7.** Distribution of *D. houi* and *C. funebris* suitable habitat under future climate scenarios.

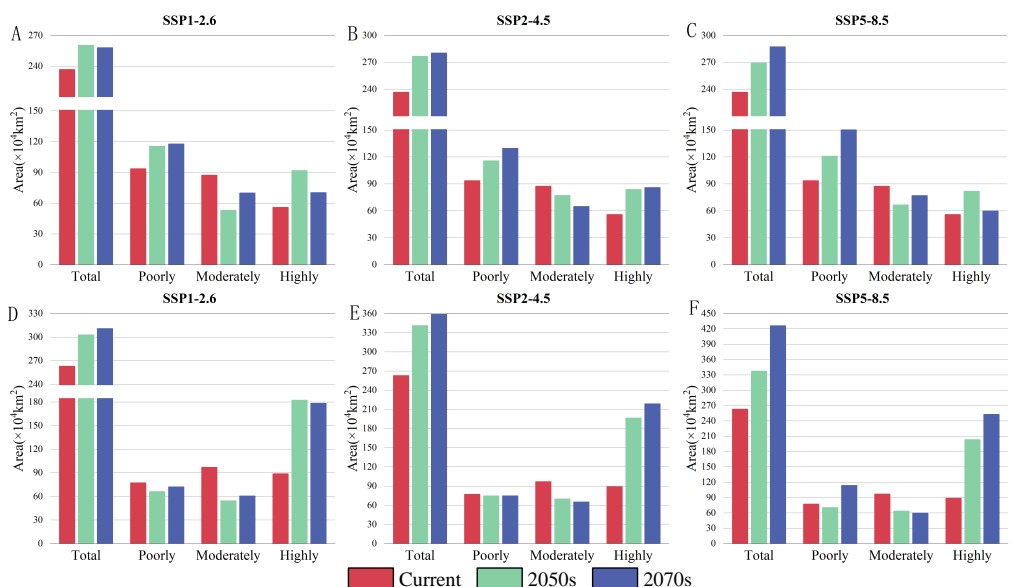

**Figure 8.** Changes in the area of suitable habitat for *D. houi* and *C. funebris* under current and future climatic conditions. In the figure, "Total" represents the total area of suitable habitat, "Poorly" represents the poorly suitable habitat, "Moderately" represents the moderately suitable habitat, "Highly" represents the highly suitable habitat. Figures (**A**–**C**) show statistics on the area of suitable habitat for *D. houi*, and Figures (**D**–**F**) show statistics on the area of suitable habitat for *C. funebris*.

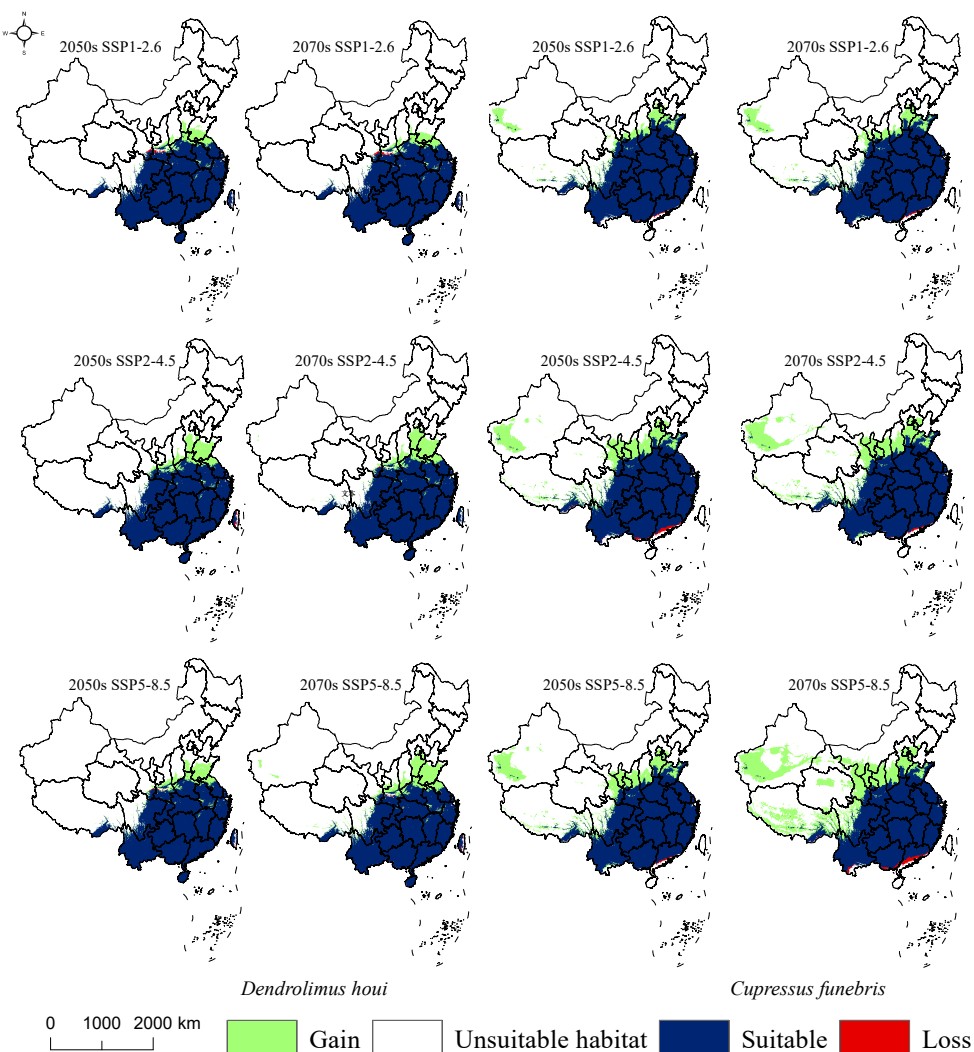

**Figure 9.** The changes in the distribution of suitable habitats of *D. houi* and *C. funebris* under different climate scenarios.

*3.5. The Shift of Suitable Habitat Center*

The current center of *D. houi*'s habitat is located at 110°7′ E, 27°50′ N in Hunan Province. Under different climate scenarios, the center of *D. houi*'s habitat exhibits varying shifts (Figure 10): In the SSP1-2.6 climate scenario, by the 2050s, it moves northeast to 110°30′ E, 28°23′ N, and by the 2070s, it shifts southwest to 110°24′ E, 28°18′ N. In the SSP2-4.5 climate scenario, by the 2050s, the center moves northeast to 110°39′ E, 28°53′ N, and by the 2070s, it moves north to 110°39′ E, 28°57′ N. In the SSP5-8.5 climate scenario, by the 2050s, the center shifts northeast to 110°35′ E, 28°40′ N, and by the 2070s, it continues northeast to 110°29′ E, 29°4′ N.

As for *C. funebris*, the current center of its habitat is situated at 110°25′ E, 29° N in Hunan Province. Under different climate scenarios, the center of *C. funebris*' habitat undergoes significant shifts: In the SSP1-2.6 climate scenario, by the 2050s, it moves from 109°21′ E, 29°55′ N to 108°50′ E, 30°8′ N and, by the 2070s, it is at 108°1′ E, 30°50′ N. In the SSP2-4.5 climate scenario, by the 2050s, it shifts from 108°4′ E, 30°35′ N to 105°8′ E, 31°51′ N. In the SSP5-8.5 climate scenario, the center moves significantly northwestward from 108°4′ E, 30°35′ N in the 2050s to 105°8′ E, 31°51′ N in the 2070s. It is noteworthy that *D. houi* and *C. funebris* share a common pattern in the direction of migration: possible migration to higher latitudes.

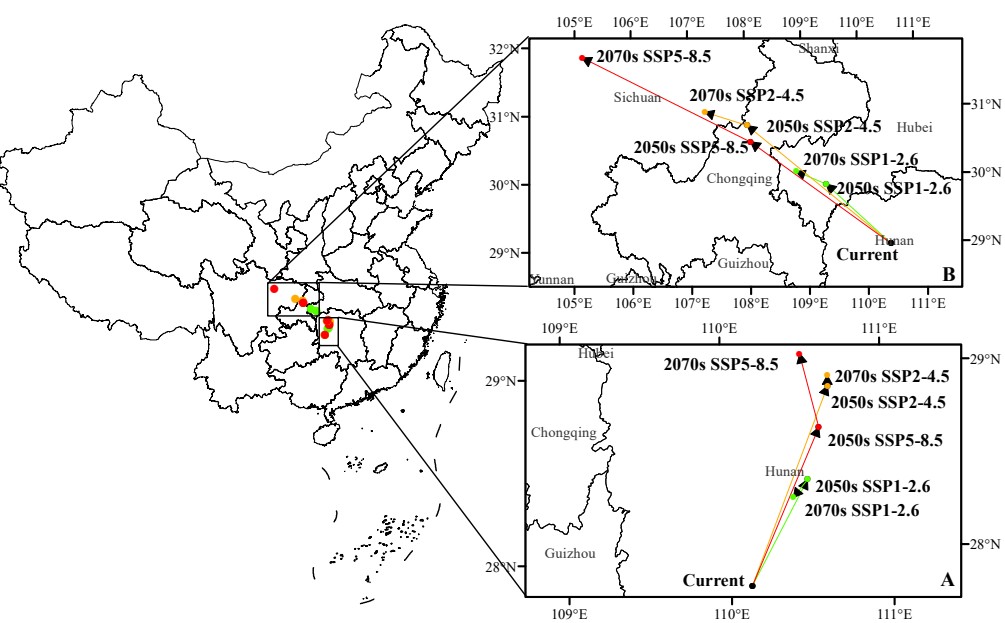

**Figure 10.** The shift of (**A**) *D. houi* and (**B**) *C. funebris* suitable habitat center. The arrows in the figure indicate the direction of potential habitat suitability center migration.

### 3.6. Changes in Intersecting Regions

*C. funebris*, as one of the most suitable host plants for *D. houi*, exhibits a distribution pattern that closely aligns with the suitable habitat of *D. houi*. The spatial distribution of the suitable habitat for *D. houi* and *C. funebris* is depicted in Figure 11. The regions of potential infestation where their habitats intersect are primarily situated in parts of Tibet, Yunnan, Sichuan, Chongqing, Guizhou, Guangxi, Hunan, Guangdong, Jiangxi, and Fujian, while Xinjiang, Qinghai, Gansu, Shaanxi, Shanxi, and other regions are less affected by *D. houi*.

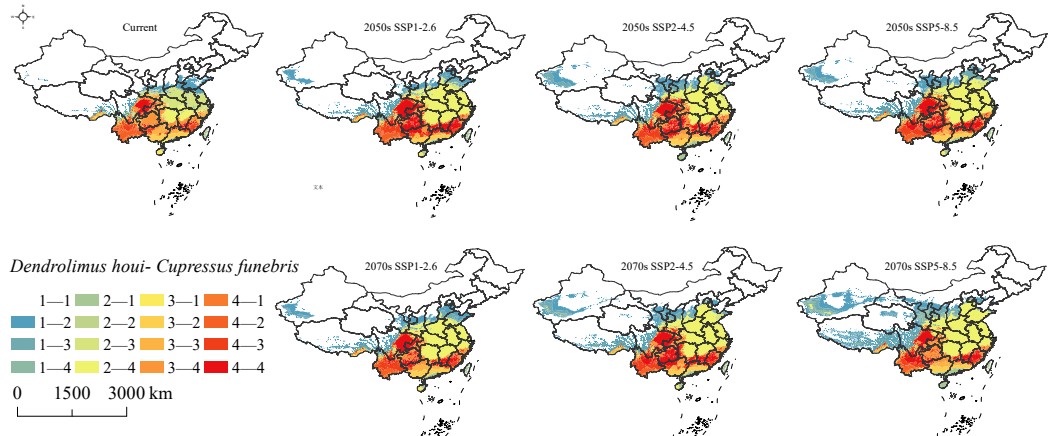

**Figure 11.** Changes in the suitable habitat distribution of *D. houi* and *C. funebris* intersections under different climate change scenarios. ("1" indicates unsuitable habitat, "2" indicates poorly suitable habitat, "3" indicates moderately sitable habitat, "4" indicates highly suitable habitat, the number before the hyphen represents the potential habitat suitability level for *D. houi*, and the number after the hyphen represents the potential habitat suitability level for the *C. funebris*).

## 4. Discussion

### 4.1. Reliability of Simulation Results

In this experiment, the Kuenm data package was employed to optimize the selection of MaxEnt model parameters. Based on the analysis of AUC evaluation criteria, the AUC

values for both *D. houi* and *C. funebris* prediction models using MaxEnt exceeded 0.9, indicating an excellent model performance. This suggests that the predicted suitable habitat distribution of *D. houi* and *C. funebris* aligns well with the suitable habitat class. The outcomes of this study hold significant reference value for research on the spatial geographical distribution of *D. houi* and *C. funebris*.

*4.2. Influence of Climatic Factors*

The impact of global warming on insects has been extensively documented, with rising temperatures observed across much of the Northern Hemisphere [32]. Additionally, future droughts in China have been projected [33]. Dale et al. conducted experiments confirming that elevated temperatures and drought can enhance the fitness and abundance of the tree pest, *Melanaspis tenebricosa* [34]. Tang et al. used the MaxEnt model to simulate the suitable habitat distribution of *Bursaphelenchus xylophilus* and its hosts, demonstrating that global warming favors the activity of *Bursaphelenchus xylophilus* vectors and significantly intensifies the damage caused by pine wood nematode disease [35]. The environmental response curves generated by the MaxEnt model illustrate the relationship between changes in environmental variables and the probability of species occurrence. Presence probabilities exceeding 0.5 indicate that such environmental conditions represent key features of a highly suitable habitat for the species. The contribution of environmental factors indicated that temperature was the primary factor affecting the suitable habitat distribution of both *D. houi* and *C. funebris*. For *D. houi*, the presence probability exceeded 0.5 in conditions such as the Min Temperature of Coldest Month (Bio6) ranging from 0.1 to 10 °C, the Temperature Seasonality (Bio4, Temperature Seasonality is the ratio of the monthly average temperature to the standard deviation of the monthly average temperature. An increase in the Temperature Seasonality indicates a gradual increase in temperature difference) from 141 to 648, and of Isothermality (Bio3) from 44 to 57. Similarly, for *C. funebris*, presence probability exceeded 0.5 in conditions including a Min Temperature of Coldest Month (Bio6) from −2 to 5 °C, a Temperature Seasonality (Bio4) from 573 to 841, and an Isothermality (Bio3) from 14.5 to 28.5. Areas where environmental conditions within the suitable habitat match these ranges may be more susceptible to significant pest outbreaks (Figure S1). The majority of highly suitable areas for *C. funebris* and *D. houi* are situated to the south of the Qinling-Huaihe River, characterized by mild winters and low precipitation. Specifically, the average temperature in January exceeds 0 °C, aligning with the temperature conditions conducive to the highly suitable habitats for both species. The intersection analysis results of *C. funebris* and *D. houi* suitable areas indicate potential large-scale infestations in the middle subtropical and northern subtropical regions. The central subtropical region typically experiences abundant annual precipitation and maintains a mean annual temperature of 16–20 °C. The mean temperature of the coldest month ranges from 5–10 °C, ensuring a relatively warm winter. Given that *D. houi*'s egg period spans from mid-September to mid-April of the following year [36], the coldest month's temperature in these areas is favorable for the normal growth and development of *D. houi* eggs, contributing to the occurrence of large-scale infestations [37]. Under the SSP2-4.5 climate scenario, the highly suitable area for *D. houi* and *C. funebris* extends to most of the Hanzhong Basin in the northern subtropics, characterized by mild winters and low rainfall. This climate facilitates the smooth growth and development of *D. houi*. The climatic characteristics of the central and northern subtropics align with the environmental conditions required during *D. houi*'s egg stage. Therefore, under a warming climate scenario, these temperature zones become crucial areas for control measures.

*4.3. Changes of Suitable Habitat*

The total suitable habitat area for both *D. houi* and *C. funebris* steadily increases with rising greenhouse gas concentrations, peaking in the 2070s. Over time, *D. houi*'s suitable habitat expands, raising the risk of large-scale infestations. It is crucial to identify suitable time frames for proactive human intervention and prevention measures. The majority of

the high habitat area for *D. houi* is concentrated in southwest China, aligning with the distribution of its host plant. Without effective prevention and control measures, the potential for unforeseeable forest disasters in this region becomes a significant concern. The suitable habitat for *C. funebris* continues to expand, with distribution occurring everywhere except in the northeast region. Under the SSP5-8.5 climate scenario for the period 2061–2080, it is projected to reach $252.75 \times 10^4$ km$^2$. This expansion presents an opportunity for ecological construction in the Northwest region, landscaping initiatives, and the cultivation of *C. funebris* as a timber species. Regions such as Xinjiang, Qinghai, Gansu, Shaanxi, and Shanxi, which experience less severe damage from *D. houi*, could be designated as protected areas for introducing and planting *C. funebris*. The stable and suitable habitats of both *D. houi* and *C. funebris* exhibit a high degree of overlap. Analyzing the direction of the center of mass movement reveals a shared tendency for expansion towards the north. Therefore, special attention should be given to the protection of *C. funebris* resources in the northern regions. Considering that *D. houi* has numerous host plants, the potential exists for increased pressure on these host plants over time in the northern regions.

The analysis of intersecting areas of *D. houi* and *C. funebris* (Table S1) reveals a declining trend in the area of unsuitable habitat where both species can simultaneously survive in China. This reduction is expected to reach its minimum during the period of 2061–2080 under the SSP5-8.5 climate scenario, with a decrease of 24.44% compared to the present period. This suggests that climate change in China does not significantly impact the coexistence of *D. houi* and *C. funebris*. Moreover, the expanding unsuitable habitat for *D. houi* and the highly suitable habitat for *C. funebris* indicate that, in the absence of *D. houi*'s influence, favorable climate conditions in China play a crucial role in the expansion of *C. funebris*'s suitable habitat. The high suitability zones for both *D. houi* and *C. funebris* have shown varying degrees of expansion compared to the present period ($19.76 \times 10^4$ km$^2$), potentially transformed from regions with medium suitability for *D. houi* and high suitability for *C. funebris*. Notably, the SSP1-2.6 and SSP5-8.5 climate scenarios both peak during the period of 2041–2060, reaching $58.14 \times 10^4$ km$^2$ and $56.52 \times 10^4$ km$^2$, respectively. In the SSP2-4.5 climate scenarios, the suitable habitat areas for *D. houi* and *C. funebris* gradually increase during the periods of 2041–2060 and 2061–2080, with the latter period tripling compared to the present. This suggests a potential risk of a large outbreak of *D. houi* in the period from 2061 to 2080. Therefore, it is essential to prioritize *D. houi* control efforts during the 2041–2060 period to prevent extensive infestations.

*4.4. Implications for Conservation Planning*

Our study offers a comprehensive prediction and analysis of the current and future potential distribution of *D. houi* and *C. funebris*. We have thoroughly explored the influence of climate and altitude factors on the distribution of these species, identifying key environmental factors. We found that the three crucial climatic factors, namely Isothermality (Bio3), Temperature Seasonality (Bio4), and Min Temperature of Coldest Month (Bio6), simultaneously impact the potential distribution of *D. houi* and *C. funebris*. This discovery has significant implications for *D. houi* control and the conservation of *C. funebris* resources. It underscores the necessity of considering climate factors in pest control and the conservation of tree resources. Our results also reveal a shift in the suitable habitat centers of *D. houi* and *C. funebris* toward higher latitudes in the future. While *C. funebris* demonstrates a slight difference by moving towards higher latitudes and altitudes, *D. houi* favors high-latitude areas as its expansion grounds. To prevent the rapid expansion of *D. houi*, it is crucial to enhance monitoring, prevention, and prediction efforts in the host forests located in high-altitude areas. Simultaneously, high-latitude and high-altitude regions offer an opportunity to introduce *C. funebris* as a favorable tree species for local development, benefiting timber production, ecological construction, and environmental beautification. Our study also identifies highly suitable habitat areas where *D. houi* and *C. funebris* intersect, primarily located in Tibet, Yunnan, Sichuan, Chongqing, Guizhou, Guangxi, Hunan, Guangdong, Jiangxi, and Fujian. The period from 2041 to 2060 may witness a potential outbreak of *D. houi*.

Therefore, we recommend prioritizing the protection of *C. funebris* forests in these regions, allocating more funds and resources for their preservation. This approach could have a positive impact on reducing the damage caused by *D. houi* infestations. In conclusion, our study advances our understanding of the potential distribution of *D. houi* and *C. funebris* and contributes to the conservation of biological resources.

## 5. Conclusions

This study employed the optimized MaxEnt model to simulate the current spatial distribution of *D. houi* and *C. funebris*, explore the crucial environmental factors influencing their distributions, simulate suitable habitat distributions under future climate change scenarios (SSP1-2.6, SSP-2.5, SSP5-8.5) for the 2050s and 2070s, and conduct intersection analyses. Findings indicate that temperature plays a pivotal role in influencing the potential distribution of *D. houi* and *C. funebris*. As a consequence of climate warming, their distribution is anticipated to expand towards higher latitudes and altitudes. Future climate change will expand the area of potential habitat for *D. houi* and *C. funebris*. Furthermore, *D. houi* is likely to cause the most severe damage to *C. funebris* between the years 2041 and 2060. Timely preventive and control measures should be implemented during this period. This study not only predicts the potential future occurrence trends of *D. houi* but also offers a theoretical foundation for the effective control of this pest.

**Supplementary Materials:** The following supporting information can be downloaded at: https://www.mdpi.com/article/10.3390/f15010162/s1, Figure S1: Response curves of 7 environmental variables in a suitable habitat distribution model of *D. houi* and *C. funebris*; Table S1: *D. houi* and *C. funebris* intersection area statistics.

**Author Contributions:** G.M.: Conceptualization, Methodology, Software, Validation, Investigation, Data Curation, Writing—Original Draft, Writing—Review and Editing, Visualization; Y.Z.: Conceptualization, Methodology, Software, Writing—Review and Editing, Supervision; Y.W.: Writing—Review and Editing; C.Y.: Writing—Review and Editing; F.X.: Validation, Supervision; Y.S.: Conceptualization, Supervision; Y.C.: Conceptualization, Resources, Writing—Review and Editing, Supervision. All authors have read and agreed to the published version of the manuscript.

**Funding:** This research was funded by National Natural Science Foundation of China grant number 31960142, 61962055.

**Data Availability Statement:** If you need this part of the experimental data, you can send an email to mgt@swfu.edu.cn to obtain it.

**Conflicts of Interest:** The authors declare that they have no known competing financial interests or personal relationships that could have appeared to influence the work reported in this paper.

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
