# Peer review of "Suitable Habitat Prediction and Analysis of Dendrolimus houi and Its Host Cupressus funebris in the Chinese Region"

_forests, doi:10.3390/f15010162_

Round 1

Reviewer 1 Report (Previous Reviewer 1)

Comments and Suggestions for Authors

This paper has been improved compared to the previous version, but there are still careless mistakes such as inconsistencies in cited references and grammar mistakes. I recommend a thorough check once again.

L10-12

These information (regularization multiplier and feature combination) were too specific and difficult to understand without explanations. I recommend to delete this sentence from Abstract.

L14

Please clarify what this 60% refers to.

L25

Citation is necessary for this description.

L38 "production and development of C. funebris [4]"

Reference 4 is not related to this topic.

L41 "biological control and monitoring [5]"

Replace [5] with [4] [5].

L65-80

I recommend to delete this paragraph including an equation. This description doesn't help readers to understand what is MaxEnt. If you want to explain the MaxEnt method, you should first explain what entropy is. Explanation for MaxEnt is sufficient in the following paragraph.

L89 "assessment of species invasions [18] [19]"

Reference 19 is not related to this topic.

L90 "medical plant studies [20]"

Replace [20] with [19].

L90 "conservation of endangered species [21] [22]"

Replace [21] [22] with [20] [21].

L106

Citation is necessary for Kuenm data package.

L116

Citation is necessary for ArcGIS.

L120-125

These sentences are missing verbs.

L128

Citation is necessary for ENMTools.

L135

Citation is necessary for WorldClim.

L138

Cite a reference 22 for CMIP6.

L145-146 "Handling multicollinearity  ... [24] [25]"

Reference 24 is not related to this topic.

L148-149, L150-151

These sentences are missing subjects.

Table 1

Please cite this table at appropriate paragraph.

L153-156

These sentences are missing subjects.

L159-168

These descriptions doesn't help readers to understand "regularization multiplier" and "feature combination". While "linear" and "quadratic" are understandable, "hinge", "product" and "threshold" are incomprehensible without explanations.

L190-191 "The iteration run type was set to "Subsample","

You should explain what is "Subsample".

L193 "is commonly used"

What is important is not whether the method is commonly used, but whether the method is used in this study.

L222-225

These are repetitions of Materials and Methods.

L227 "threshold (H)" -> "threshold (T)"

L227 "AIC value of 0"

Table 2 shows "delta AICc", not "AIC". Delta signifies an increment, therefore, it shows the increment in AICc as compared to AICc of the optimized model.

Table 2

Title is difficult to understand the meaning. Please revise the explanation for delta.AICc.

L294-300

This should be described in Discussion.

L302-308

Cite Figure 10 in this paragraph.

L315-324

This should be described in Discussion. Here authors discussed the reason why the future distribution of D. Houi and C. funebris was predicted to expand northward. Authors discussed "possibly seeking areas with more ...", "responding to climate change by moving ..." and "This suggests its search for new suitable areas ...", but It is not possible to interpret the results of the model in this way. It is okay to interpret the result of a real shift in the distribution of the target species in this way, however, you predicted the future suitable areas based on the climate change scenarios. Future suitable areas should be managed carefully because of its high probability of the pest migration, but the migration is not yet occurred.

L335-341

These should be described in Materials and Methods.

L345-346 "emphD. houi" -> "D. houi"

L370-372 "for C. funebris. For D. houi" -> "for C. funebris and D. houi"?

L374 "Intersection analysis results of C. funebris."

Is this the title of some part?

Comments on the Quality of English Language

I recommend English proofreading. Please refer to comments for details.

Author Response

Dear Reviewer:

Many thanks for handling our manuscript entitled “Suitable habitat prediction and analysis of Dendrolimus houi and its host Cupressus funebris in the Chinese region” (Manuscript Id: forests-2788032) and giving us the opportunity to revise the manuscript again.

Your comments and suggestions has been carefully considered, and the paper has been revised accordingly. All rectifications to the manuscript in accordance with your comments has been highlighted in Microsoft Word.

We have rephrased the problematic parts to the best of our ability. We hope that these revisions can meet the reviewers’ expectations.

Best regards,

Guangting Miao

Reviewer 2 Report (Previous Reviewer 2)

Comments and Suggestions for Authors

Dear colleagues, 

I have got not any remarks to your manuscript after revision. 

Best wishes! 

Author Response

  Thank you for your previous suggestions for our articles, it helps a lot to improve the quality of the articles, thank you very much. Wish you the best!

Reviewer 3 Report (Previous Reviewer 3)

Comments and Suggestions for Authors

Dear Editor,

After re-reading the submitted article entitled "Suitable habitat prediction and analysis of Dendrolimus houi and its host Cupressus funebris in the Chinese region", I must note that all my comments have been corrected and the content of the manuscript has been supplemented with missing information. Therefore, I see no arguments against accepting this article in its current revised form for publication in the journal FORESTS.

Author Response

  Thank you for your previous suggestions for our articles, it helps a lot to improve the quality of the articles, thank you very much. Wish you the best!

This manuscript is a resubmission of an earlier submission. The following is a list of the peer review reports and author responses from that submission.

Round 1

Reviewer 1 Report

Comments and Suggestions for Authors

This paper predicts the suitable habitat of Dendrolimus houi and its host Cupressus funebris using species distribution model based on their current distribution. Future expansion of their distribution was also predicted by the model. My major concern is the way authors interpret the results of MaxEnt model. Materials and methods is too redundant, and many sentences have no subject. In contrast, figure legends are too short in general. Some results are described in Discussion. Most part of Conclusion is the summarization of results and not necessary. Authors should first arrange the format of the paper properly.

Major comments

1. Interpretation of MaxEnt results

Based on the results of MaxEnt model, authors discussed that "This expansion suggests that the climate of the Chinese region is becoming increasingly favorable for the survival and proliferation of these species." (L357-359) and "This suggests its search for new suitable areas at higher latitudes and altitudes" (L398-399). MaxEnt model includes the climatic variables and the future distribution of the insect and its host was predicted by this model using the global warmings scenarios. Therefore, it is natural that the suitable area will expand and that the area shift northward. Authors should distinguish what can be and what can't be discussed from the results.

2. Materials and methods

Authors first summarize their methods in L92-152, and this is not necessary. Figure 1 can be cited at the last of Materials and methods section. In that case, please correct the miss spelling such as "ENMToos", "Regularizatio" and "tese". In L153-273, many sentences start from verb and lack subject. English of this section should be checked by a native speaker. This part looks like manual because of the frequent use of the term "your" and "you". Generally, the explanation is too redundant. For example, descriptions such as L164-167, L182-186, L197-200, L219-222 are not necessary. Please avoid the redundant explanations, and clearly describe what you did in this study.

3. Figures and Tables

In general, figure legends are too short. Each figures and tables should be self explanatory. You should at least explain the meaning of each axis of the graph. Table 1 is difficult to understand because there are no explanations for "RM", "FC" and "delta AICc". In addition, L, Q, P, H are the abbreviations and should be explained.

4. Results and Discussion

Figures 7-11 are included in Discussion. These are the results of the application of MaxEnt model, therefore, should be included in Results section. Correspondent descriptions should also be moved to Results section. Therefore, most part of the sections 4.1, 4.2, 4.3 and 4.4 should be included in Results section.

5. Conclusion

Most part of Conclusion section is the summarization of results. Please avoid redundancy and clearly state what can be concluded from this study.

Specific comments

Title

Please italicize the scientific names.

L11-13

It is difficult to understand "regularization multiplier" and "Q, T, and H" without explanation.

L26-27

Please use a normal font for author's name and family and order name.

L29-30

Please cite references for the current distribution of this pest.

L32-33 hm2 -> km2

Please delete "square kilometers".

L56-58

Please italicize the genus name.

L76-79

Please italicize the scientific names.

L229-230

What is important here is whether you use this index AICc or not in this study.

L290-291

L304-305

L315-317

Suggestion from results should be included in Discussion.

L326-331

Please italicize the scientific names.

L377-399

Please properly cite Figure 10 in these paragraphs.

Comments on the Quality of English Language

English of this paper, especially Materials and methods section, should be improved.

Reviewer 2 Report

Comments and Suggestions for Authors

Dear collegues, 

I have got minor remarks only: 

- The abbreviations Q, T and H in abstract (l. 13) must be clarified in the same place or removed; 

- You must italicize the latin names of genes and species (see l. 26, 56, 58, 76, 78, 79, 147, 221, 222, 241, 251, 254, 260, 327-330); 

- You must not italicize any other words in the plain text (e.g., order and family names in l. 27 or measure unit in l. 32); 

- The part of text in l. 74-79 (from "Notably" to "industry") looks excessive. I guess, the above-mentioned examples are enough for illustration of your argumentation; 

- The map on Figure 2 is too small and the points upon it are poorly visible; 

- The correlation matrix on Figure 3 is uninformative without labels. Your readers could be interested in the correlation structure of factors, but You didn't sign which factors You had compared. The codes (bio01, bio02 etc) could not help with this. Completely the same concerned to y-axis labels in Figure 5; 

- Some environmental factors names have got obvious sense, but other are not. E.g., the meaning of 'seasonality' term is not clear. You should give the explanations in 2.2 or another section; 

- The legends in Figures 7–9, 11, codes of scenarios in Figure 10 can be read with difficulty only. You should make their font size bigger. It is possible if leave one legend per figure only (for 7-9); 

-  The statements in l. 411-413 should be proved by citations. 

I hope my remarks are useful for You.

Best wishes!

Reviewer 3 Report

Comments and Suggestions for Authors

The thesis submitted to me for review entitled "Suitable habitat prediction and analysis of Dendrolimus houi and its host Cupressus funebris in the Chinese region" deals with a very important topic: habitat analysis and prediction of the occurrence of Dendrolimus houi and its host Cupressus funebris in the Chinese region. The paper was prepared very carefully and most of the descriptions fully cover the topic. It is commendable that the authors have modeled the future distribution of both the insect pest and the host in different time periods of the 2050s (2041-2060) and 2070s (2061-2080), considering three future climate change scenarios. However, the biggest concern I have is related to the relatively small number of data sets (57) on the occurrence of D. houi. Taking into account the division into training and validation groups (75 – 25) mentioned by the authors, the models are based on only 42-43 occurrence points of D. houi. Nevertheless, I would like to emphasize that the results obtained are still very interesting and certainly deserve to be published. Below are some minor comments that I think need to be added:

- Consider the scientific name when changing Dendrolimus houi (D. houi) for the first time for clarity, and make sure to capitalize the genus and species names (e.g. Dendrolimus houi).

- Please provide more detail on the economic losses caused by D. houi and the ecological impact on host plants (e.g. C. funebris).

- Briefly explain how the MaxEnt model works, especially for readers who are not familiar with this model.

- Provide a summary of the entropy-based principle and how it is applied in modeling species distribution.

- Provide a rationale for the selection of ENMTools for data refinement.

- Explain the rationale for selecting 1160 candidate models.
